# Implementation of Green Infrastructure in Existing Urban Structures: Tracking Changes in Ferencváros, Budapest

**Gabriel Silva Dantas** [1,*] , **Ildikó Réka Báthoryné Nagy** [1] **and Pedro Brizack Nogueira** [2]

1    Department of Urban Planning and Urban Green Infrastructure, Institute of Landscape Architecture Urban Planning and Garden Art, Hungarian University of Agriculture and Life Sciences, 1118 Budapest, Hungary; bathoryne.nagy.ildiko.reka@uni-mate.hu
2    Department of Exact Sciences, State University of Feira de Santana, Feira de Santana 44036-900, Brazil; peunogueira@gmail.com
*    Correspondence: dantassgabriel@gmail.com; Tel.: +36-30-113-3678

**Abstract:** Understanding the resilience of urban forms as a latent force that drives a place's physical characterization and social cohesion is essential for defining successful adaptive processes of pre-existing urban fabrics. Budapest's ninth district (Ferencváros) is an outstanding example of transforming a complex historical urban context, which underwent renovation strategies guided by maintaining and enhancing essential morphological elements. Courtyards have great relevance in conditioning the well-being in areas of high occupational density, especially in terms of accessibility to urban green infrastructure. In the case of Ferencváros, they were reframed to add new layers of use and to improve territorial integration by unifying smaller private courtyard unities into more extensive communal areas, creating a comprehensive urban green network, preserving urban heritage, and increasing green coverage. This study assesses how this recent re-urbanization phenomenon is related to political changes in a post-socialist city. The conjuncture found in Ferencváros is unique, yet it can be applied in other similar contexts. The methodology applied to this study is supervised classification for the quantitative analysis of remote-sensing image data with GIS software assistance—a procedure rarely applied in medium-scale urban analysis. However, it was verified to be precise and effective in tracking morphological changes. The preliminary results indicate a significant intensification in greenery in the urban pattern, especially in the core areas of the blocks: the courtyards. After the intervention, green areas became more predominant, cohesive, and articulated.

**Keywords:** courtyard; urban green infrastructure; urban pattern; urban renewal; supervised classification



## 1. Introduction

More than two-thirds of the population of the countries of the European Union currently live in urban areas. Despite the growing effort to include the urban development agenda as a mainstream element of the bloc's Cohesion Policy, it was only in the period between 2007 and 2013 that the EU established it as one of its main development guidelines. At first, the European Commission began to investigate the problems to a conceptual degree and act on an experimental basis, which later also resulted in intergovernmental cooperation agreements in the field of urban development in favor of cohesion [1].

On a global level, this was a response to a major paradigm shift: the advent of the intensification of the globalization process, which boosted the strengthening of international ties, aiming to achieve economic and social prosperity. However, the trend towards globalization occurs unequally in different contexts. The developing world countries would have more opportunities to access exponential growth, while facing difficulties in income distribution and expanding the city's infrastructure network. Around the world, in Asian and South American cities, or even in post-Soviet centers, two eminent development patterns can be identified: peripheral growth (both office spaces and houses) and the participation of

the private sector in infrastructure provision [2]. Despite the similarities, the socioeconomic particularities of each region outlined the conditions for the occurrence of this process. Little has been studied regarding the reverberation of this phenomenon in post-socialist urban centers, especially concerning adaptive measures for their inclusion in a global urban network [3].

This scenario was also asymmetric among the countries of the European Union, which motivated the intensification of isolation and urban segmentation [1]. Post-Soviet countries, such as Hungary, only joined the political–economic bloc later, and faced challenges left by the communist legacy, such as deficiencies in urban infrastructure, management of mass-housing estates, and environmental problems [4].

With the end of the communist regime, the urban centers previously inserted in this socio–spatial conjuncture went through an initial population shrinkage, marginalization, and isolation [5]. Public policies played a fundamental role in structuring the privatization operation and introducing the market-based economy. Cities underwent, therefore, an acute adaptive procedure through a situation of disturbance of the political and social order [6]. On the one hand, it was essential to create mechanisms to ensure urban resilience, both in the socioeconomic scope and morphologically, by preserving urban patterns and social aspects, making cities more attractive and competitive [7]. On the other hand, investments in green infrastructure to promote urban resilience did not follow the same tendency in most eastern European cities. The case of Ferencváros stands out in this context, especially for the centralized coordination between public and private interests to rebuild a sizeable urban stretch.

Environmental issues are one of the most significant drawbacks to urban development, considering the interdependence between social, economic, and environmental dimensions for achieving sustainable urban development [8]. In cities with infrastructural limitations, such as Budapest, the obstacles in promoting social cohesion are even more significant. For this reason, mitigating accessibility to green infrastructure can be a tool for territorial integration and a consequent increase in urban resilience [9].

Vienna is often an analogous example at the urban-structure level when establishing a comparative framework with Budapest. These cities share numerous similarities due to their intertwined historical processes. Despite the gradual disconnection they have gone through since World War II, recently, both have faced a common issue: urban decay and increasing urban voids [10]. Notwithstanding their divergent political conditions, between 1970 and 1990, several projects targeting the renovation of central areas emerged in those cities.

One of the biggest obstacles in observing changes in cities is the lack of detailed and model-ready morphological data at the urban scale. In Vienna, GIS data analysis proved to be efficient in obtaining the morphological heterogeneity across the urban landscape, which implies the possibility of using this method to track changes in the infrastructure of Budapest [11].

The IX District of Budapest is an example of the requalification of an emptied historic area with heterogeneous territorial occupation, marked by the existence of urban voids (initially occupied by small- and medium-sized industries) [12]. The voids generated by restructuring the regional industrial production system contributed to the acceleration of the urban decay process and territorial fragmentation, making it difficult for the population to remain in the area [13]. Historic urban morphology, traditionally found in eastern central European cities, is also found in Ferencváros. Before the intervention, most buildings had typological characteristics such as continuous and aligned facades and individual courtyards [14].

In this intervention, the restructuring of the courtyard system was essential for implementing an extensive system of green infrastructure in the region, shaping the intensification of green areas in the densely occupied urban fabric. This action took into account morphological elements characteristic of the site and had a relevant impact on increasing urban cohesion.

## 2. Materials and Methods

The investigation follows a case-study approach, relying on research on evidence of satellite image processing and map analysis to identify variations in the morphological structure and the green infrastructure over time. The methodology was drawn on qualitative and quantitative analysis of land-use and land-cover (LULC) mapping, highlighting the transformations that occurred in the polygonal of the study in historical periods remarkable in terms of variations in the pattern of urban development [15]. The analysis was carried out in three different periods, in 2000, 2011, and 2021, in order to obtain parameters to establish a comparative framework. The input dates were defined based on milestones in the change in conduct in the management of the urban domain and migration movements—especially regarding Ferencváros—and the availability of material for investigation.

The study area comprises a specific region within the entirety of the IX District of Budapest and is located between the geographic coordinates 47°28′48.82″ north latitude and 19°04′38.91″ east longitude and has an extension of 71.5403 hectares. The database used is composed of digital images from orbital sensors made available by Google Earth Pro on its image catalog by NASA, and were processed in an IACS-compatible environment, on geographical information system (GIS). The software used in this analysis was ArcGis version 10.1, using the ArcToolBox tools extension, with components designed for supervised classification such as Create Sign-nature, Filter statistics, and Maximum Likelihood Classification [16].

The classes selected to carry out the supervised classification consider artificial and natural elements that constitute the urban landscape. Land-cover classes are identified as natural earth resources (e.g., forests, water, bogs, marshlands), while land-use classes are considered artificial areas (e.g., agriculture, roads, cities) [15]. The classes used for the supervised classification are vegetation, exposed soil, street, and building. These classes are essential to identify and follow the pattern of urban development in the region, highlighting demolitions, new buildings, streets and pedestrian paths, urban voids, and green areas over time. Twenty sample pixels for each of the classes mentioned before were obtained, thus enabling measurement on different dates.

Due to the relatively low resolution of the images accessed to perform the procedure, the results found present a degree of inaccuracy. In addition to that, clouds and shadows are expected in optical remote-sensor images, decreasing the precision of the analysis. The occlusion of features is another limiting factor, which reduces the available useful area of the image, compromising the quantitative analysis [17]. Those factors also made it unmanageable to obtain images for classification before the 2000s, which were freely available, and with a satisfactory resolution.

Although some urban areas, mainly in Asia and North America, have been the object of scientific study with supervised classification, little material is produced on this subject in Eastern European cities. Furthermore, this methodology is commonly found in studies covering large territorial portions, often aiming to measure the growth of urban areas towards rural or natural environments, for example [18]. In the case of this present case study, the supervised classification is applied to identify changes in the urban morphology in a segment of the ninth district of Budapest, generating evidence of the renovation operation performed in this region.

## 3. Results

The Ferencváros requalification project differs from other initiatives in Budapest in this field, as it was set up to enhance urban resilience through the maintenance and improvement of morphological elements of structural importance to the urban fabric, such as the original layout of blocks, continuous facades, overall height, and plot size (or use of architectural elements that symbolize the original individuality of each of them, even when multiple sites were joined for the development of new development) [19]. However, the courtyards are the most striking feature in the definition and spatial articulation in the region. These elements were restructured, enabling new layers of use.

The courtyards, initially independent, were primarily devoid of greenery or had poor and fragmented green areas with restricted access to the building's residents [20]. The restructuring proposed the unification of these elements, also encompassing the urban voids and creating a mesh of public or semi-public pedestrian crossings. The LULC maps produced from satellite images were used in this study as a tool to define the impacts of this intervention on spatial conformation.

### 3.1. Social–Political Conjuncture and the Designation of Input Satellite Images

The first restructuring projects of Frencváros date from the late 1970s and were conceived under modern guidelines of spatial elaboration and articulation, foreseeing the demolition of most pre-existing historic buildings and emphasizing the creation of new roads, green areas, and buildings for residential and community use [16]. The project underwent revisions in the 1980s, making it more adaptable to local circumstances, as shown in Figure 1. The decision to reconfigure the city's outskirts emerged from the increasing densification of its agglomeration belt, with the migratory movement from the countryside to the city, accelerating the demand for rapid housing construction [21]. Despite the extensive planning, few changes were implemented in the area during this period.

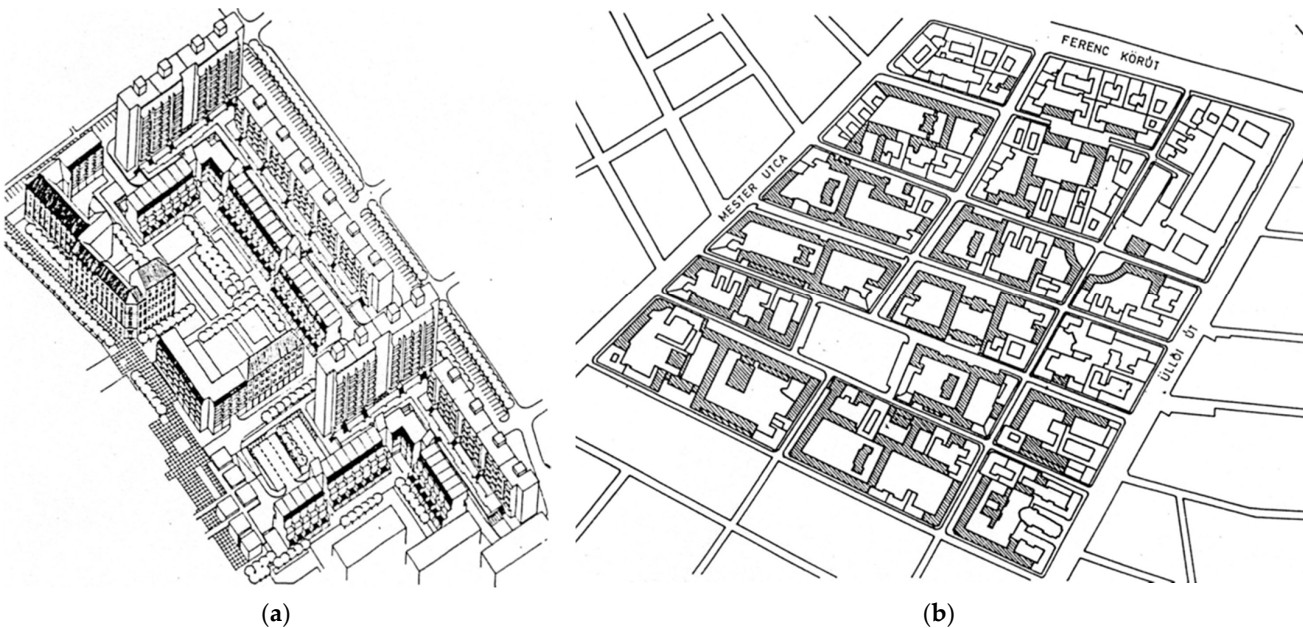

(**a**)  (**b**)

**Figure 1.** (**a**) Scheme of a requalification plan from the late 1970s for the Middle Ferencváros. Most of the historical buildings were set to be demolished and replaced by large prefab residential blocks [10]; (**b**) Revised master plan from 1982 to 1983. The original street network and the layout of the urban blocks are redesigned to bear wider streets and more extensive communal courtyards. For the first time, the intention to implement a public square was indicated in the south–central region of Middle Ferencváros. New buildings are hatched [14].

The political transition process in Hungary began in the early 1990s, but it was only around the 2000s that national and international investments in real estate and public infrastructure expanded [22]. The process of urban sprawl and suburbanization started in the late 19th century. This escalated in the mid-20th century in Budapest and impacted the densification of the Ferencváros region, a trend that continues to rise. Table 1 [16] indicates that the urban sprawl movement grew significantly between 2001 and 2011.

**Table 1.** Continuous urban sprawl process in Budapest even after the communist era.

| Year/Region | 1990 | 2001 | 2011 |
|---|---|---|---|
| Proper Budapest | 2,016,000 | 1,775,000 | 1,729,040 |
| Agglomeration Belt | 567,000 | 672,000 | 805,848 |
| Total Agglomeration | 2,583,000 | 2,447,000 | 2,534,888 |

Furthermore, as mentioned earlier, between 2007 and 2013, the EU established urban development as one of the most relevant factors for its cohesion policy. This decision boosted more evident transformations in the city's urban structure, especially in areas with significant transformative potential and subject to real estate investments, as occurred in the analyzed district [23]. For the reasons mentioned, the year 2011 was selected as one of the landmarks for the elaboration of this research. Finally, the third period selected for sampling was the year 2021, seeking to obtain more recent data that portray the current situation found in the place.

*3.2. Urban Design and the Implementation of Green Infrastructure*

At the local level in Budapest, the end of the communist regime implied the return to self-governance in Budapest, accordingly inferring the two-tier administrative system and the subsequent shift in decision-making from the city to the district level. This scenario of increasing the individuality and competencies of the districts has resulted in their ability to reformulate their social and housing policies, making them capable of launching urban requalification projects [24].

In the global policies scope, the set of new Sustainable Development Goals (SDG), established in the post-2015 Development Agenda in September 2015, is defined in the General Assembly of the United Nations [25]. Among the guidelines mentioned in the document, SDG 11.7 specifies the following for public spaces:

> "*By 2030, provide universal access to safe, inclusive and accessible, green and public spaces, particularly for women and children, older persons and persons with disabilities*".

Regarding accessibility to green and leisure areas, Ferencváros presented deficiencies. The coefficients of accessibility to those areas in the district until the 2000s (when urban intervention was not yet consolidated) were below 9 m$^2$ per capita within 15 min of walking distance stated by the World Health Organization [26]. To achieve these parameters, the system of integrated green courtyards was designed, in addition to the implementation of green elements at specific points on the streets—a measure that made the core and the edges of the blocks greener, also improving the landscape conditions of the place.

To meet the indicators of the World Health Organization and to achieve a better ratio between population density and accessibility to green open spaces, a new compact urban park [27] was created from the demolition of some poorly conserved buildings and empty plots in the south–central region of the polygonal, as shown in Figure 2. The Kerekerdő park is one of the most significant public elements for the configuration of the green grid in the area, being a (public) confluence point for the green paths [28]. The same is true of Ferenc tér, a pre-existing square in the north–central region that also plays the role of a significant urban green infrastructure element.

Among the benefits achieved by implementing a comprehensive green system are strengthening social relations, increments in connectivity, improvements in urban cohesion, and an increase in local economic activities, leading to the resilience of this urban territory [9]. Ensuring accessibility to green open spaces is a possible response to a healthier urban environment [29]. In that perspective, the presence of green spaces, pedestrian paths, and leisure equipment was accomplished by redesigning former brownfields and residential plots.

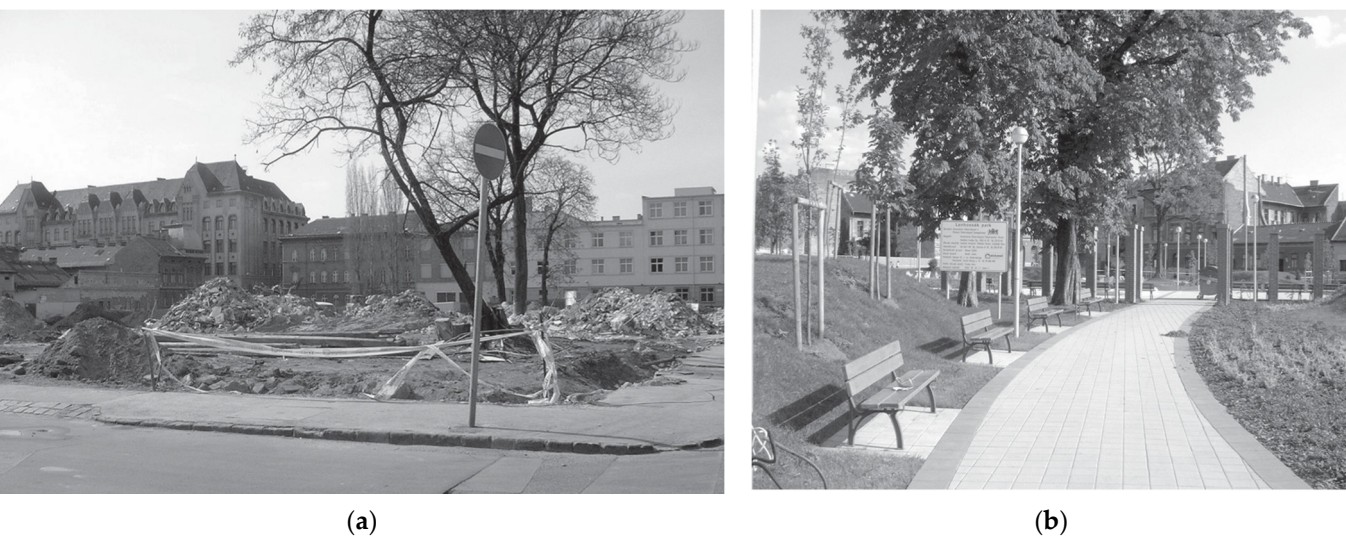

**Figure 2.** (**a**) Process of establishing the Kerekerdő park from the agglutination of empty plots and the demolition of residential buildings [14]; (**b**) Kerekerdő park already in use in the early 2000s. The public space was designed to prioritize the integration of the territory and promote accessibility to green areas and playgrounds [14].

### 3.3. Supervised Image Classification for Tracking Urban Transformations

As shown in Figure 3, the supervised classification performed in the satellite image of the 2000s reveals a territory still lacking a comprehensive urban green infrastructure. The green elements are presented in a fragmented and diluted way, evidencing the typological characteristics of a historical urban fabric ascended from closed blocks and buildings developed around relatively narrow and poorly lit courtyards, devoid of green components capable of significantly impacting the landscape composition and the quality of life for local inhabitants [30].

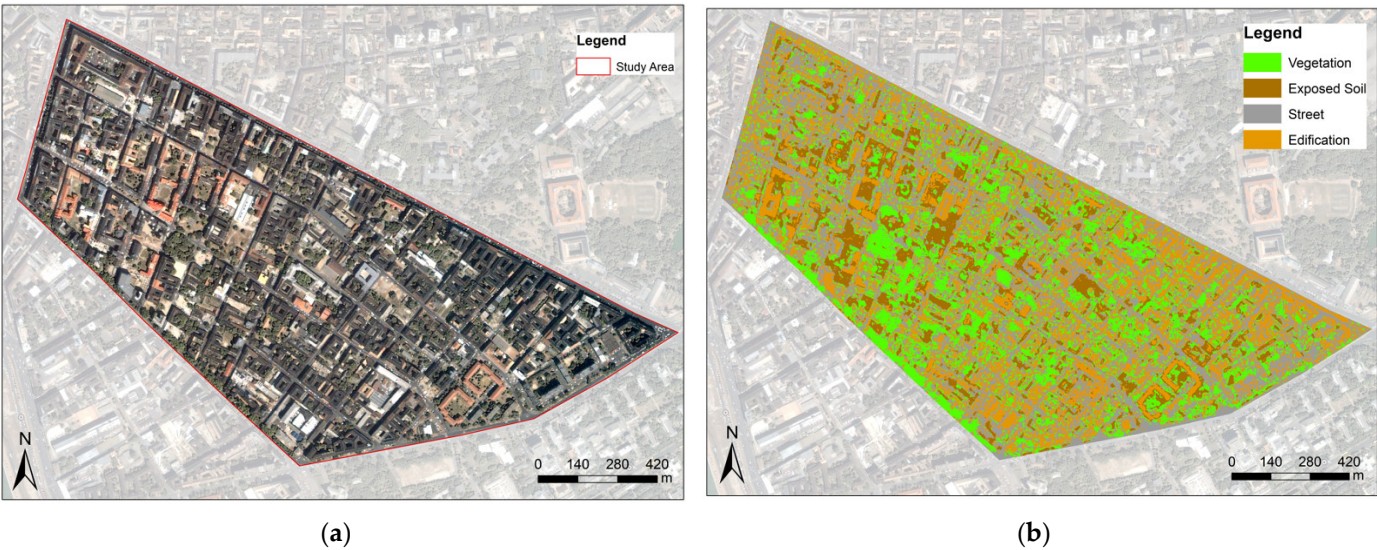

**Figure 3.** (**a**) Satellite image from the 2000s. It is possible to notice that most buildings have typological characteristics common to historic structures, which reverberate in the dense urban condition, despite the existent industrial voids.; (**b**) The supervised classification reveals urban fragmentation and the deficiency of green infrastructure in the area.

There are also high rates of exposed soil, indicating, at this time, the prevalence of urban voids characteristic of deactivated industrial facilities [12]. These idle areas were

mainly concentrated in the central portion of the study polygonal since the edges (better served by the public transport network and important mobility hubs), already in this period, were mostly occupied by residential and commercial buildings.

In 2011, it was already possible to visualize the grid of the green infrastructure defined in the urban requalification project, as indicated in Figure 4. The reduction in exposed soil areas is notorious, as are the growth of areas occupied by buildings. This period is marked by the coexistence of several new residential developments with old historic buildings in a poor state of conservation, designated for demolition, but most of which had not yet been demolished or rebuilt—except for a set of buildings located in the vicinity of Kerekerdő park [31], in the south–central region of the polygonal. In this area, there is a marked predominance of exposed soil.

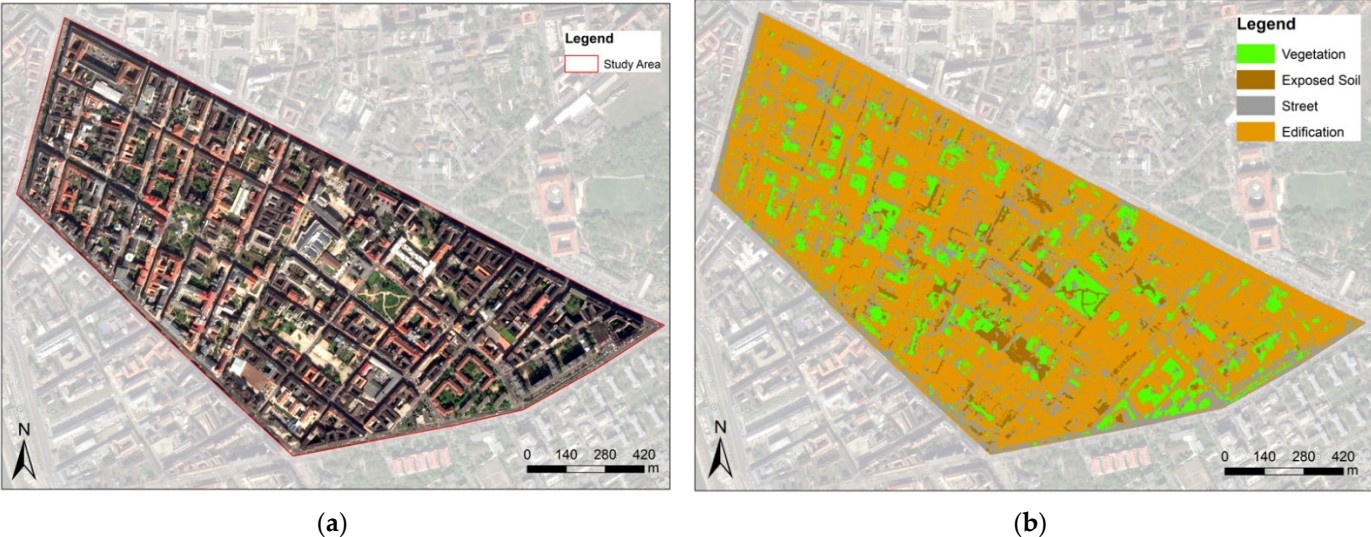

(**a**)　　　　　　　　　　　　　　　　　　　　　　　　　　　　(**b**)

**Figure 4.** (**a**) Satellite image from 2011. It is clear that the intervention has already reached a certain degree of consolidation, and the redesign of some urban blocks has already been completed, especially in the northern region; (**b**) The supervised classification indicates the first significant advances toward urban cohesion and green infrastructure begins to emerge.

Despite the prominent involvement of the private sector in the reconstruction process, this was a project guided by the demands arising from the local public administration. Contrary to most districts of Budapest after the political transition, the local government was able to continue the Middle Ferencváros renovation project, as this area was officially designated as an "urban rehabilitation site" [16]. For this reason, the stages of development of the intervention were defined primarily from the perspective of social conflicts, and aimed to address solutions to the most urgent infrastructural deficiencies. The city designated the priority areas as "centers of gravity", and the green infrastructure would establish the articulation between them. The implementation of Kerekerdő park, for example, was one of the measures adopted to stimulate the development of one of the regions of the district with the most significant deficit in infrastructure, landscape conditions, and socio–spatial cohesion [31].

Based on the 2021 satellite image analysis, visible in Figure 5, it is possible to observe the consolidation of the greenspace system. The intensification of linear and compact green elements, implemented in previous stages of the intervention in the planned urban fabric, is noted. At the same time, simultaneously, new shared courtyards were also established. At this stage, the proportion between green components and built area is more balanced than in previous stages of the requalification process, with approximately twice as many areas occupied by buildings as green areas, as indicated in Table 2.

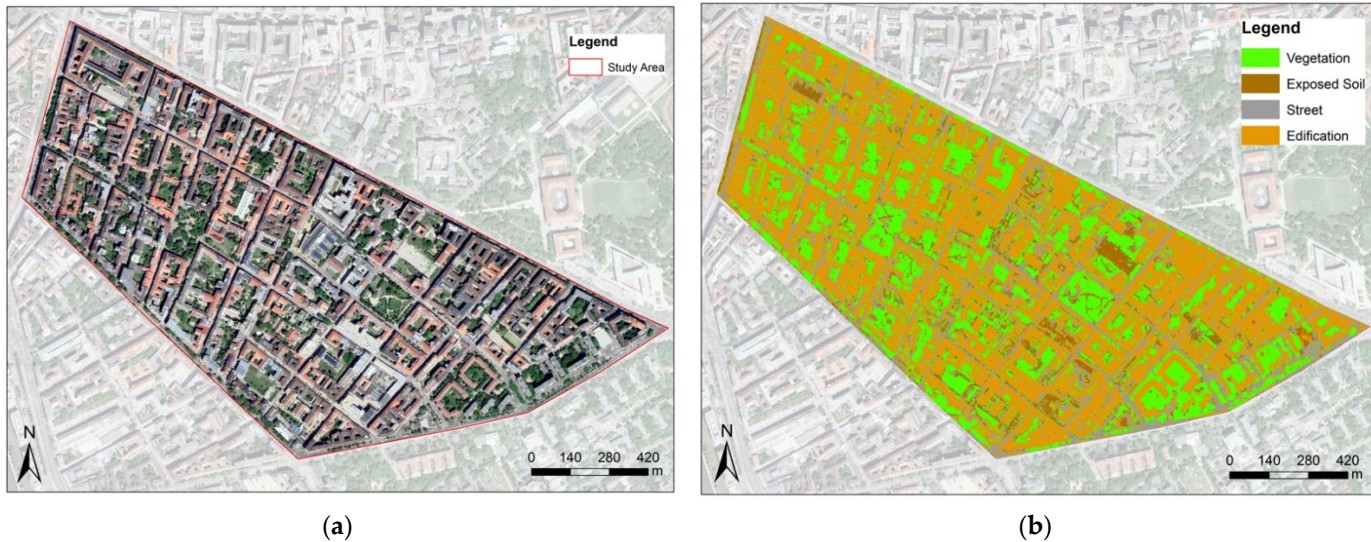

**Figure 5.** (**a**) Satellite image of 2021. Based on the morphological analysis, it is clear that the intervention has reached its stage of maturity; (**b**) implementation of green infrastructure is almost complete, and there are few areas marked as exposed soil.

**Table 2.** The total area occupied by each of the respective classes over time. Results were obtained with supervised classification.

| Class Name | 2000 | 2011 | 2021 |
|---|---|---|---|
| Exposed soil | 9,193,263 | 3,964,979 | 5,005,173 |
| Street | 26,787,694 | 13,280,933 | 11,864,784 |
| Building | 25,055,177 | 51,467,673 | 41,295,866 |
| Vegetation | 17,627,973 | 10,004,571 | 20,546,853 |

The green areas are also more equidistantly distributed throughout the territory, bringing the conditions currently existing in the district closer to the parameters established by the World Health Organization concerning accessibility to urban public green areas [26]. However, there is still a denser strip of occupation where the urban green infrastructure system still did not reach its full potential, east of Kerekerdő park. This is one of the last "centers of gravity" among the priority areas for the requalification implementation. There is still a relatively high percentage of historic buildings that have not undergone renovation or demolition interventions in this area.

## 4. Discussion

Changes in the social–political context of post-socialist cities caused challenges in terms of their ability to persevere as resilient urban forms, leading them to the need to adapt to the new model of social and spatial structuring. In this context, international public policies, mediated by the European Union, were designed to assist in the development of these cities, aiming at cohesion and reducing the gap—especially in terms of infrastructure and environmental management—that exists between eastern and western European cities [6].

Parallel to these events, the dynamics of suburbanization and urban sprawl also guided cities such as Budapest to a scenario of social and physical fragmentation (also in terms of landscape conditions), particularly on the fringes of large metropolitan areas such as the situation found in Ferencváros [32]. In this sense, green infrastructure is an essential component in the viability of sustainable urban planning, facilitating social cohesion and supportive social networks, enhancing equity, and the development of social capital and promoting a healthier environment [33].

The partial or total demolition of some historic buildings during the renovation process raises questions about urban heritage preservation and urban morphology. In that regard, Kropf (2018) [34] states the following:

> *"Configuration is an arrangement of parts, and a type is a configuration with a degree of modularity and integration as a cultural habit. The type is a configuration that is or has been actively reproduced. While each example of a type might be slightly different, the configuration remains the same."*

Following that perception, demolition performed in some specific points of the district and the reconfiguration of the historic courtyards, despite being antagonistic to the preservation of urban heritage and its morphological characteristics, allowed for the revival of the area's configuration. This process happened through the typological re-constitution, providing cohesion to the fragmented urban territory and boosting its resilience [35].

The results reported in this paper show that the urban requalification of Ferencváros was efficient in improving territorial cohesion through a gradual reduction in urban voids (represented mainly by the exposed soil class) and the implementation of an extensive system of green elements. It is relevant to underline that the year 2011, as an intermediate period of evolution of the urban condition, presents classes with percentages of land use and occupation different from the general trend seen between 2000 and 2021 (Table 2). This movement is justified by the fact that this was a transition period, pointing to the coexistence of sections already restructured with others where the action was still in progress. The co-consolidation of the transformations was only confirmed in the outcome of the investigation carried out in the most recent image.

Furthermore, the supervised classification of satellite images has proved to be a robust method for analyzing the gradual reconstruction activity. The LULC maps produced with the help of GIS tools provide a detailed report on the development of the different layers that combined composed the urban fabric over time in the narrowly selected periods [36]. Nevertheless, the relatively low resolution of the images used for the study resulted in higher misclassification rates because of the spectral similarity, mainly between the exposed soil and the building materials. The use of higher resolution images (produced by Sentinel-2, for example) and the employment of Synthetic Aperture Radar (SAR) to extract textural features could be possible solutions to this issue [37].

As Hammerberg et al. (2018) demonstrated in Vienna, the use of remote-sensing imagery to determine morphological features on an urban scale can be beneficial for tracking urban structure and precisely evaluating the performance of cities at the environmental level [11]. The classes analyzed in the study proved to be substantially sufficient in the stratification of land-use information to capture the heterogeneity of the urban landscape morphology. Similar results were reported by Forget et al. (2018) through the supervised classification based on manually digitized training samples to frame the heterogeneity of sub-Saharan African urban areas. The automated approach could reach classification effectiveness similar to that of manual sampling strategies [37]. In this case study, the authors identified some inaccuracies related to the attributes of the openly available aerial images (difficulties also found in the study developed in Ferencváros).

The supervised classification as a method assisted in the correlation of the development strategies adopted by the public–private initiative with the remarkable historical events that defined the project's guidelines. Nevertheless, the impossibility of obtaining older images of free access conditioned the analysis to periods in which the intervention had already started, limiting the comparative framework concerning previous periods.

**Author Contributions:** Conceptualization, G.S.D.; methodology, G.S.D.; software, G.S.D. and P.B.N.; validation, G.S.D. and I.R.B.N.; formal analysis, G.S.D.; investigation, G.S.D.; resources G.S.D.; data curation, G.S.D.; writing—original draft preparation, G.S.D.; writing—review and editing, G.S.D.; visualization, G.S.D. and P.B.N.; supervision G.S.D. and I.R.B.N.; project administration, G.S.D.; funding acquisition, I.R.B.N. All authors have read and agreed to the published version of the manuscript.

**Funding:** This research received no external funding and the APC was funded by the Hungarian University of Agriculture and Life Science.

**Informed Consent Statement:** Not applicable.

**Data Availability Statement:** Publicly available datasets were analyzed in this study. This data can be found here: [https://earth.google.com/] (accessed on 10 March 2022).

**Conflicts of Interest:** The authors declare no conflict of interest.

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
