# Peer review of "Implementation of Green Infrastructure in Existing Urban Structures: Tracking Changes in Ferencváros, Budapest"

_land, doi:10.3390/land11050644_

Round 1
Reviewer 1 Report
The article presents an interesting topic of regeneration of urban tissue based on a green system. In part of the introduction, it is recommended to deepen the references to world research. The method and subject of the research are properly described. There are gaps in the specification of data sources (e.g. 173). The article is largely descriptive and the research method only documents the changes made in the urban space. It is recommended to deepen the work towards supra-local references.
Author Response
Dear reviewer,
Thank you very much for the comments.
As required, new parallels were established with supra-local references, mentioned in the introduction and discussion, and emphasized the global scenario in general.
We chose to focus our comparative frameworks with Vienna - because of the numerous morphological similarities and historical connections with Budapest - and with a study developed in African cities. In the second, we found strong correspondence with the methodology applied.
We hope to have found satisfactory solutions for the mentioned issues.
Kind regards,
Gabriel Dantas.
Reviewer 2 Report
The manuscript draws on the technical means of remote sensing and GIS research to track changes in urban structure. There are some concerns about this manuscript, both in terms of its innovation and its presentation. The authors first need to address the following issues:
- What is the innovation of this manuscript?
- Abstract section: The results and new findings need to be emphasized rather than spending a lot of space on the background.
- Introduction section: It needs to be supplemented with existing studies related to the research content of this manuscript. It is also necessary to highlight the research gap by summarizing and reviewing others' studies to emphasize the innovation and significance of this study. For example, are there any studies that have been conducted to track changes in urban structure with the help of image data? What are the analytical perspectives of the existing studies? What are the differences between the analytical perspectives of the study in this manuscript and the existing studies?
- Materials and Methods section: This section is not rigorous. The situation of satellite remote sensing data used in the manuscript is not described. The process of supervised classification is not elaborated and described in detail, and the authors are requested to add it. In addition, how are the results validated after supervised classification? What is the validation accuracy? Please add the experiments and related explanatory notes.
- Results section: The tables and figures in the manuscript are not clearly indicated in the elaboration of the resulting text. In other words, only the figures and tables are placed in the manuscript, but which textual descriptions do the figures and tables correspond to in the article? Please clarify and label appropriately.
- Discussion section: The discussion section is slightly thin.There is a lack of in-depth thinking and a lack of comparison between this study and the results of existing studies.Please add it carefully.
The manuscript was not titled Conclusion, Please check the content and structure of the manuscript carefully.
Author Response
Dear reviewer,
Thank you very much for the detailed correction and comments.
As requested, the introduction has been restructured to highlight the results found.
Regarding the originality and innovation of the paper, although some urban areas, mainly in Asia and North America, have been the object of scientific study with supervised classification, little material is produced on this subject about Eastern European cities. Furthermore, this methodology is more commonly found in studies covering large territorial portions, often aiming to measure the growth of urban areas towards rural or natural environments. Nevertheless, our approach focus on medium-scale urban land-use and land-cover analysis.
New parallels were established with supra-local references, mentioned in the introduction and discussion, emphasizing the global scenario.
We chose to focus our comparative frameworks on Vienna because of the numerous morphological similarities and historical connections with Budapest and a case study developed in African cities. In the second, we found strong correspondence with the methodology applied.
The methodology section has been expanded with new references dealing with accuracy, resolution, and occlusion aspects.
The results and the references were restructured and provided links to avoid confusion when reading the data.
We opted not to add conclusions, as this section is described as optional in the template provided by the journal and is recommended in cases where the "discussion is unusually long or complex."
We hope to have addressed satisfactory solutions for the issues.
Kind regards,
Gabriel Dantas.
Reviewer 3 Report
The article is very interesting and clearly contributes to the better understanding of the urbanization of the Hungarian capital.
However, the title reflects more the classification methodology than the problematic itself, which is the implementation of an extensive system of green elements in the observed districts. A title that more accurately states the issue, which is however well mentioned in the discussion, is needed.
Please indicate the cities where the authors' Universities are located
It would probably be necessary to use aerial images from the years before 2000 or at least from 1990 onwards in order to understand the land use evolution since the communist era. If these images are grayscale, use grayscale image segmentations when possible. This would be a better baseline to support the discussion.
In the classification it seems that there is confusion between edification /exposed soil and roads which makes it difficult to compare real differences between 2000 and 2011. Explanations are missing on how the shading of the images is taken into account in the classifications. The authors could also compare the quality of the classifications according to several methods.
Author Response
Dear reviewer,
Thank you very much for the detailed correction and comments.
Indeed, the title did not adequately reflect the issues discussed in the manuscript, so it was replaced accordingly.
As required, the authors' respective cities have been added.
Unfortunately, it was not possible to locate satellite images before the 2000s, freely available and with a satisfactory resolution to perform the supervised classification. We are currently acquiring new images with the financial support of our educational institutions to develop further analysis in the future. We are aware of this limitation.
As noted in your review, some study classes present relatively high misclassification rates due to the spectral similarity, mainly between the exposed soil and the building materials. This problem is related to the images' low resolution and is now also reported in the manuscript.
We hope to have addressed satisfactory solutions for the issues.
Kind regards,
Gabriel Dantas.
Round 2
Reviewer 3 Report
This revised version can be published. Thanks to the authors for the modifications and answers to questions.